# IRGS++: Robust Geometry, Material, and Light Decomposition with Accelerated Inter-Reflective Gaussian Splatting

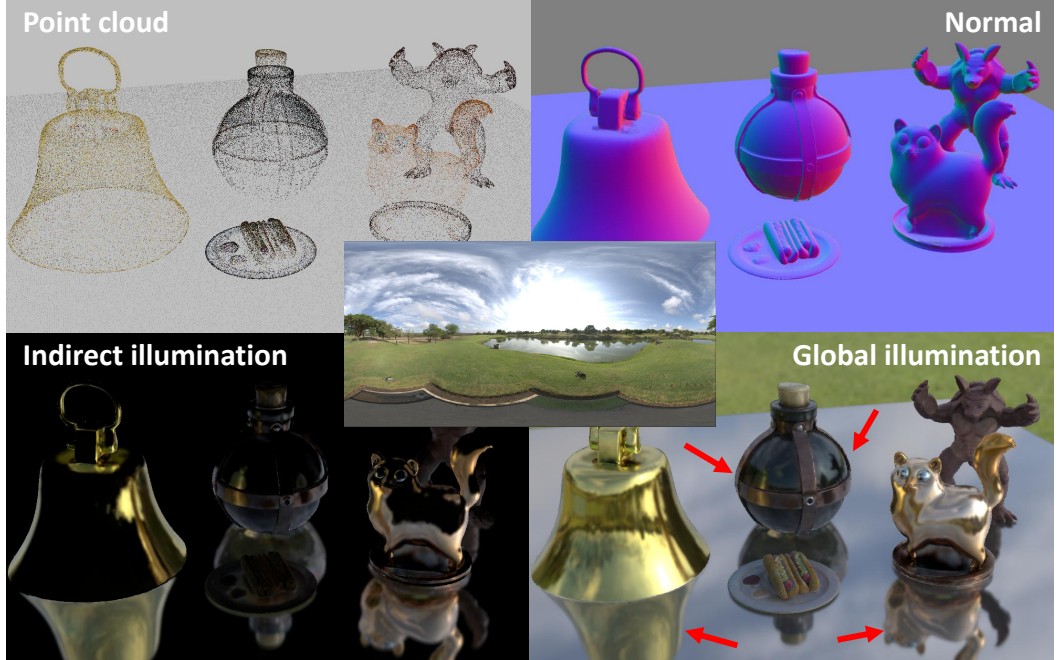

Figure 1: **IRGS++** demonstrates photorealistic secondary illumination in relit scenes containing both low-gloss and glossy surfaces, achieving plausible light effects with only 32 rays per pixel.

## Abstract

The accurate evaluation of the rendering equation is a fundamental challenge in inverse rendering, as it governs the modeling of complex light-surface interactions. Existing 3DGS-based methods face a key trade-off: approaches using split-sum approximations fail to model secondary light effects, while those relying on heavy Monte Carlo integration suffer significant rendering slowdowns. To address this, we present *IRGS++*, an accelerated inter-reflective Gaussian splatting framework for inverse rendering that effectively handles both low-gloss and glossy materials. To reduce ray sampling in Monte Carlo integration, we implement multiple importance sampling with distinct distributions (cosine, GGX, and light sampling) to better capture light effects. We also apply a cross-bilateral filter to the Monte Carlo estimator, reducing noise while preserving quality with limited ray samples. Furthermore, we replace 2D Gaussian ray tracing with mesh-based ray tracing during relighting, cutting per-ray computations from hundreds of ray-splat checks to a single ray-triangle intersection. Extensive experiments demonstrate *IRGS++*'s superior performance among 3DGS-based competitors on both low-gloss and glossy datasets while achieving a 50-fold acceleration over IRGS.

## 1 INTRODUCTION

Inverse rendering is a long-standing problem in computer vision and computer graphics. It decomposes scene attributes, geometry, material, and lighting from multi-view images. This decomposition enables realistic relighting of reconstructed scenes by applying estimated materials under novel illumination. Recent advances in scene representations, notably neural radiance fields (NeRF) Mildenhall et al. (2020), which encode 3D scenes as continuous volumetric fields parameterized by MLPs, and 3D Gaussian splatting (3DGS) Kerbl et al. (2023), which models scenes as collections of 3D Gaussians, offer substantial opportunities to advance inverse rendering pipelines.

NeRF-based methods Zhang et al. (2021); Liu et al. (2023); Hasselgren et al. (2022) use neural implicit representations with ray marching to model materials and light effects. Nvdiffrec-MC Hasselgren et al. (2022) improves upon Nvdiffrec's Munkberg et al. (2022) split-sum approximation by adding multiple importance sampling (MIS) and a differentiable denoiser, which reduces ray samples while keeping rendering quality. However, these methods still require heavy computation due to neural network queries. The advent of 3D Gaussian splatting, known for real-time rendering and high-fidelity reconstruction, has driven significant inverse rendering innovations Liang et al. (2024); Zhu et al. (2024b); Gao et al. (2024); Gu et al. (2025a). Existing 3DGS-based methods mainly adopt two strategies: The first category Liang et al. (2024); Zhu et al. (2024b) simplifies rendering equations to avoid costly Monte Carlo integration. GS-ROR$^2$ Zhu et al. (2024b) combines split-sum approximation with learned signed distance fields (SDFs) to better model reflective surfaces, though this simplification inherently limits accurate modeling of complex light transport. The second category Gao et al. (2024); Gu et al. (2025a) implements full rendering equations with extensive Monte Carlo sampling. IRGS Gu et al. (2025a) introduces stratified sampling and 2D Gaussian ray tracing to accurately capture ray visibility and radiance for inter-reflections, but requires intensive per-pixel ray sampling and underperforms on glossy surfaces.

In this work, we extend IRGS Gu et al. (2025a) to better model specular surfaces and accelerate rendering while preserving relighting fidelity. Inspired by Nvdiffrec-MC Hasselgren et al. (2022), we integrate variance-reduction techniques, multiple importance sampling (MIS) and image denoising, into 3DGS-based inverse rendering. Our proposed *IRGS++* framework delivers two advances: (1) material modeling from diffuse to highly specular surfaces, and (2) substantial rendering acceleration over IRGS Gu et al. (2025a). We implement MIS for the rendering integral using three sampling distributions: cosine-weighted, GGX Heitz (2018), and the environment light distribution Pharr & Humphreys (2010). Following Nvdiffrec-MC, we apply a cross-bilateral filter Schied et al. (2017) to final renders to suppress Monte Carlo noise while preserving fidelity at low sample counts. For relighting, we replace IRGS's 2D Gaussian ray tracing with mesh-based ray tracing, reducing per-ray work from hundreds of ray-splat intersections to a single ray-triangle intersection and achieving an order-of-magnitude speedup. Importantly, instead of using 2DGS Huang et al. (2024) for geometric initialization, we adopt Ref-Gaussian Yao et al. (2025) to better capture specular surfaces.

Extensive experiments demonstrate *IRGS++*'s efficacy in handling both low-gloss and glossy objects. *IRGS++* achieves 50-fold acceleration compared to IRGS Gu et al. (2025a), and establish new state-of-the-art relighting performance in 3DGS-based inverse rendering pipelines. In Figure 1, we visualize the point cloud, normal, indirect illumination, global illumination of a relit scene composited of several low-gloss and glossy objects, showing *IRGS++*'s remarkable inter-reflection effects.

The contributions of this work include: **(i)** *IRGS++*, an inverse rendering framework capable of accurately modeling both low-gloss and glossy surfaces while achieving significant rendering acceleration compared to IRGS. **(ii)** Variance reduction strategies, including multiple importance sampling and image denoising, to improve efficiency while maintaining photorealistic fidelity. **(iii)** Mesh-based ray tracing during relighting, achieving substantial reductions in computational complexity.

## 2 RELATED WORK

**Novel view synthesis.** NeRF Mildenhall et al. (2020) represents a major breakthrough in novel view synthesis by employing multi-layer perceptrons (MLPs) and volume rendering to learn continuous volumetric representations. Subsequent studies have built upon this foundation through multi-resolution hash grids Müller et al. (2022), voxels Sun et al. (2022); Yu et al. (2021), and tensor decomposition Chen et al. (2022), significantly accelerating training and rendering speeds

while reducing computational demands. Despite their effectiveness, NeRF-based methods Barron et al. (2021; 2022; 2023) remain computationally intensive, requiring long training periods and substantial resources. In contrast, 3D Gaussian splatting Kerbl et al. (2023) (3DGS) demonstrates superior efficiency by explicitly representing scenes as learnable 3D Gaussians and employing tile-based rasterization. It has been widely adopted for geometry reconstruction Huang et al. (2024); Yu et al. (2024), dynamic scene modeling Yang et al. (2023a; 2024), inverse rendering Gao et al. (2024); Liang et al. (2024), 3D generation Tang et al. (2023); Yi et al. (2024), street scene applications Yan et al. (2024b); Chen et al. (2023), and robotics Yan et al. (2024a); Ji et al. (2024). The rasterization-based framework of 3DGS, however, limits its ability to simulate ray-based optical effects. 3DGRT Moënne-Loccoz et al. (2024) addresses this through a differentiable Gaussian ray tracer that computes radiance along ray paths. We follow IRGS Gu et al. (2025a), which implements 2D Gaussian ray tracing for precise ray-splat intersections, enabling realistic inter-reflections.

**Inverse rendering.** Inverse rendering aims to reconstruct geometry, material attributes, and lighting conditions from multi-view RGB images. Many NeRF-based Srinivasan et al. (2021); Boss et al. (2021); Yao et al. (2022); Zhang et al. (2023); Verbin et al. (2022); Boss et al. (2022); Attal et al. (2025); Liu et al. (2023); Jin et al. (2023); Yang et al. (2023b); Zhang et al. (2021; 2022); Wu et al. (2024a); Liang et al. (2023); Munkberg et al. (2022); Hasselgren et al. (2022); Liu et al. (2023); Zhu et al. (2024a); Gu et al. (2025b) methods employ ray marching and neural implicit fields to address complex optical effects. Nvdiffrec-MC Hasselgren et al. (2022) integrates multiple importance sampling (MIS) Veach & Guibas (1995) with a differentiable denoiser to improve rendering efficiency while preserving quality. However, such methods remain inefficient due to long training and rendering times and limited quality. Recent approaches have applied 3DGS to inverse rendering Liang et al. (2024); Gao et al. (2024); Shi et al. (2023); Wu et al. (2024b); Guo et al. (2024); Gu et al. (2025a); Zhu et al. (2024b); Lai et al. (2025); Sun et al. (2025); Chen et al. (2025), leveraging its representational capacity by assigning material-related properties to individual Gaussian primitives. GS-ROR$^2$ Zhu et al. (2024b) utilizes a signed distance field (SDF) to supervise Gaussian geometry and adopts deferred splatting for rendering. However, its reliance on split-sum approximation oversimplifies the rendering equation, compromising material and lighting estimation accuracy. To achieve precise inter-reflection simulation, IRGS Gu et al. (2025a) implements the full rendering equation alongside 2D Gaussian ray tracing. However, the exhaustive stratified sampling in IRGS not only restricts computational efficiency but also underperforms on glossy surfaces. To address these limitations, we propose *IRGS++*, which employs the full rendering equation while integrating multiple techniques to minimize variance and accelerate rendering.

## 3 METHOD

In this section, we present *IRGS++*, a pipeline for geometry, material, and light decomposition using accelerated inter-reflective Gaussian splatting Gu et al. (2025a), capable of handling both low-gloss and glossy objects. We begin by introducing the requisite background (Section 3.1). Next, we detail our physically based rendering pipeline (Section 3.2). Then, we describe variance-reduction techniques (Section 3.3). Finally, we integrate mesh-based ray tracing into our framework to further accelerate relighting (Section 3.4). An overview of the pipeline is shown in Figure 2.

### 3.1 PRELIMINARY

**Gaussian splatting.** 3D Gaussian splatting (3DGS) Kerbl et al. (2023) models a 3D scene as a collection of 3D Gaussian primitives. Each primitive is defined by a center position $\boldsymbol{\mu} \in \mathbb{R}^3$ and a covariance matrix $\Sigma \in \mathbb{R}^{3 \times 3}$, with its spatial influence at a point $\boldsymbol{x}$ expressed as:

$$G(\boldsymbol{x}) = \exp\left(-\frac{1}{2}(\boldsymbol{x} - \boldsymbol{\mu})^\top \Sigma^{-1}(\boldsymbol{x} - \boldsymbol{\mu})\right). \tag{1}$$

Additionally, each Gaussian is associated with an opacity value $o \in [0, 1]$ and a view-dependent appearance $\boldsymbol{c}$ modeled via spherical harmonics (SH). For rendering, 3D Gaussians are projected onto the 2D image plane through a view transformation $W$ followed by perspective projection. The projected 2D covariance matrix $\Sigma'$ is approximated as: $\Sigma' = JW\Sigma W^\top J^\top$, where $J$ denotes the Jacobian of the perspective projection. The final pixel color $\mathcal{C}$ is computed via alpha-blending of ordered projected 2D Gaussians from front to back using: $\mathcal{C} = \sum_{i=1}^{N} T_i \alpha_i \boldsymbol{c}_i, T_i = \prod_{j=1}^{i-1}(1 - \alpha_j)$, where $\alpha_i = o_i \cdot G'(\boldsymbol{p})$ combines opacity and the projected Gaussian's contribution at pixel $\boldsymbol{p}$.

Figure 2: **Overview of *IRGS++*.** Leveraging the geometry of pretrained 2D Gaussians, we first generate material maps and geometry maps via splatting. The rendering equation is solved through multiple importance sampling. For indirect illumination, we implement hybrid ray tracing: 2D Gaussian-based during training and mesh-based during relighting. Finally, a cross-bilateral filter denoises the Monte Carlo estimator to produce the final image.

**Gaussian ray tracing.** While 3DGS achieves real-time rendering, it falls short in modeling ray-based effects (e.g., shadows and inter-reflections) due to its rasterization-based nature. To address this limitation, 3D Gaussian ray tracing (3DGRT) Moënne-Loccoz et al. (2024) proposes to apply ray tracing across 3D Gaussians. By leveraging a $k$-buffer hit-based marching technique with hardware-accelerated OptiX Parker et al. (2010) implementation, the method achieves both efficient and accurate rendering. In the meanwhile, 2D Gaussians Huang et al. (2024) demonstrate superior surface modeling compared to 3D Gaussians. Building on this advantage, IRGS Gu et al. (2025a) introduces 2D Gaussian ray tracing (2DGRT), thereby eliminating inconsistencies in ray-splat intersections inherent to 3D Gaussian primitives.

**Rendering equation.** The rendering equation Kajiya (1986) describes the interaction between lighting and surfaces over the hemispherical domain $\Omega$ defined by the surface normal $\boldsymbol{n}$:

$$L_o(\boldsymbol{\omega}_o, \boldsymbol{x}) = \int_\Omega f(\boldsymbol{\omega}_o, \boldsymbol{\omega}_i, \boldsymbol{x}) L_{\mathrm{i}}(\boldsymbol{\omega}_i, \boldsymbol{x})(\boldsymbol{\omega}_i \cdot \boldsymbol{n}) d\boldsymbol{\omega}_i \,, \tag{2}$$

where $L_o$ and $L_i$ denote outgoing radiance and incident radiance. The bidirectional reflectance distribution function (BRDF) $f$ encodes material response, parameterized by material properties.

## 3.2 RENDERING PIPELINE

**Rasterization.** We adopt physically-based deferred rendering, same as in IRGS Gu et al. (2025a), wherein Gaussians are first rasterized to generate pixel-level material maps before applying the rendering equation. It should be noted that IRGS relies on a dielectric material assumption, limiting its capacity to model highly reflective surfaces. To address this, we extend material parameterization by equipping each Gaussian with material attributes, including albedo $\boldsymbol{a} \in [0,1]^3$, roughness $r \in [0,1]$, and an additional metallic $m \in [0,1]$. The pixel-level maps can be obtained through rasterization:

$$\{\mathcal{C}, \mathcal{D}, \mathcal{N}, \mathcal{A}, \mathcal{R}, \mathcal{M}\} = \sum_{i=1}^N w_i \{\boldsymbol{c}_i, d_i, \boldsymbol{n}_i, \boldsymbol{a}_i, r_i, m_i\}, \text{ where } w_i = \frac{T_i \alpha_i}{\sum_{i=1}^N T_i \alpha_i}. \tag{3}$$

where $\boldsymbol{c}$ is the outgoing radiance modeled via SH and $\boldsymbol{n} = \boldsymbol{t}_u \times \boldsymbol{t}_v$ is the normal vector.

**Light modeling.** Leveraging the depth map obtained above, we can easily derive the surface point $\boldsymbol{x}$ for each pixel. We decompose the incident light at $\boldsymbol{x}$ into direct and indirect terms:

$$L_{\mathrm{i}}(\boldsymbol{\omega}_i, \boldsymbol{x}) = V(\boldsymbol{\omega}_i, \boldsymbol{x}) L_{\mathrm{dir}}(\boldsymbol{\omega}_i) + L_{\mathrm{ind}}(\boldsymbol{\omega}_i, \boldsymbol{x}), \tag{4}$$

where $L_{\mathrm{dir}}$ is assumed to come from distant sources parameterized by an environment map, while $V$ and $L_{\mathrm{ind}}$ are obtained through 2DGRT. Notably, we implement distinct strategies for $L_{\mathrm{ind}}$ during training and relighting phases. During training, $L_{\mathrm{ind}}$ is computed via alpha-blending of outgoing radiance $\boldsymbol{c}_i$ through 2DGRT, whereas its acquisition during relighting is detailed in Section 3.4.

**Monte Carlo sampling.** Given incident radiance, we employ importance sampling (detailed in Section 3.3) to numerically evaluate the rendering equation integral Cook & Torrance (1982):

$$\boldsymbol{c}_{\text{pbr}} = \frac{1}{N_{\text{r}}} \sum_{i=1}^{N_{\text{r}}} \frac{f(\boldsymbol{\omega}_o, \boldsymbol{\omega}_i, \boldsymbol{x}) L_{\text{i}}(\boldsymbol{\omega}_i, \boldsymbol{x})(\boldsymbol{\omega}_i \cdot \boldsymbol{n})}{q(\boldsymbol{\omega}_i)}, \tag{5}$$

where $N_{\text{r}}$ sampled directions $\boldsymbol{\omega}_i$ are drawn from proposal distribution $q$ with probability density function $q(\boldsymbol{\omega}_i)$. When the distribution of $q$ closely matches the integrand of rendering equation, the variance of the estimator is minimized, enabling comparable quality with fewer sampling rays.

## 3.3 VARIANCE REDUCTION

While IRGS Gu et al. (2025a) demonstrates remarkable relighting quality, its reliance on a high ray count per pixel with stratified sampling introduces significant computational overhead, requiring seconds per image during relighting. To address this inefficiency, we pursue variance reduction strategies that preserve rendering fidelity with substantially fewer samples. Inspired by Nvdiffrec-MC Hasselgren et al. (2022), we leverage multiple importance sampling Veach & Guibas (1995) (MIS) that strategically combines three distinct distributions (diffuse, specular, and environmental lighting), and then apply a post-processing image denoising process to further eliminate noise.

### 3.3.1 MULTIPLE IMPORTANCE SAMPLING

Multiple importance sampling Veach & Guibas (1995) (MIS) provides a methodology for combining samples from multiple probability distributions, enabling the sampling distribution to approximate the characteristics of target integrand. The estimator with balance heuristic is formulated as:

$$\sum_{i=1}^{n} \frac{1}{n_i} \sum_{j=1}^{n_i} m_i(X_{i,j}) \frac{g(X_{i,j})}{p_i(X_{i,j})}, \quad m_i(x) = \frac{n_i p_i(x)}{\sum_k n_k p_k(x)}. \tag{6}$$

To reduce variance when evaluating the rendering equation, we implement MIS with three distinct sampling strategies: 1) Cosine-weighted distribution targeting the diffuse component, 2) GGX distribution Heitz (2018) aligned with the specular lobe, and 3) Environment light distribution generated through intensity-based sampling of environment map Pharr & Humphreys (2010). The GGX sampling proves particularly effective for glossy surfaces by concentrating samples around specular reflection directions, enabling accurate estimation of specular contributions in $\boldsymbol{c}_{\text{pbr}}$ with only a few samples. Light sampling handles strong directional illumination, significantly eliminates artifacts.

### 3.3.2 DENOISING

Denoising in computer graphics enables high-quality rendering with low sample counts by reducing noise in Monte Carlo estimators, thereby enhancing the stability and efficiency of the rendering process. Typical denoising implementations employ spatial filter kernels that perform low-pass operations on noisy inputs through weighted neighborhood averaging. However, while aiming to use as few samples as possible in Monte Carlo integration, the limited sampling rate inevitably results in images with substantial noise. Inspired by Nvdiffrec-MC Hasselgren et al. (2022), we leverage the image denoising technique, using a cross-bilateral filter based on Spatio-temporal Variance-Guided Filtering Schied et al. (2017) (SVGF), which preserves geometric edges through depth and normal-aware weighting. The bilateral weighting between pixels p and q is computed as:

$$\text{Bilateral}(p, q) = e^{-\frac{|p-q|^2}{2\sigma^2}} e^{-\frac{|z(p)-z(q)|}{\sigma_z |\nabla z(p) \cdot (p-q)|}} \max(0, \boldsymbol{n}(p) \cdot \boldsymbol{n}(q))^{\sigma_n}, \tag{7}$$

where $z$, $\boldsymbol{n}$ denote the image space depth and surface normal, respectively.

## 3.4 ACCELERATED RELIGHTING WITH MESH-BASED RAY TRACING

During training, IRGS Gu et al. (2025a) computes incident radiance via Gaussian ray tracing, where each Gaussian contributes its learnable outgoing radiance $\boldsymbol{c}_i$. However, $\boldsymbol{c}_i$ becomes invalid under novel environmental lighting during relighting. To circumvent recursive sampling, we aggregate material properties through Gaussian ray tracing and apply split-sum approximation for incident radiance estimation. While optimized sampling (Section 3.3) alleviates computational load,

Table 1: Quantatitive comparison of normal, novel view synthesis, albedo, and relighting results on TensoIR dataset Jin et al. (2023). A higher intensity of the red color signifies a better result.

| Method | Normal MAE ↓ | Novel View Synthesis | | | Albedo | | | Relighting | | |
|---|---|---|---|---|---|---|---|---|---|---|
| | | PSNR ↑ | SSIM ↑ | LPIPS ↓ | PSNR ↑ | SSIM ↑ | LPIPS ↓ | PSNR ↑ | SSIM ↑ | LPIPS ↓ |
| NeRFactor | 6.314 | 24.68 | 0.922 | 0.120 | 25.13 | 0.940 | 0.109 | 23.38 | 0.908 | 0.131 |
| InvRender | 5.074 | 27.37 | 0.934 | 0.089 | 27.34 | 0.933 | 0.100 | 23.97 | 0.901 | 0.101 |
| TensoIR | 4.100 | 35.09 | 0.976 | 0.040 | 29.28 | 0.950 | 0.085 | 28.58 | 0.944 | 0.081 |
| GS-IR | 4.948 | 35.33 | 0.974 | 0.039 | 30.29 | 0.941 | 0.084 | 24.37 | 0.885 | 0.096 |
| R3DG | 5.927 | 37.34 | 0.982 | 0.021 | 26.20 | 0.913 | 0.095 | 27.37 | 0.934 | 0.064 |
| GS-ROR$^2$ | NA | NA | NA | NA | NA | NA | NA | 27.07 | 0.938 | 0.060 |
| IRGS | 3.998 | 35.52 | 0.964 | 0.049 | 33.42 | 0.954 | 0.076 | 30.63 | 0.935 | 0.076 |
| Ours ($N_r$=512) | 3.980 | 35.15 | 0.967 | 0.045 | 33.95 | 0.949 | 0.079 | 32.12 | 0.949 | 0.059 |
| Ours ($N_r$=32) | - | - | - | - | - | - | - | 31.83 | 0.944 | 0.065 |

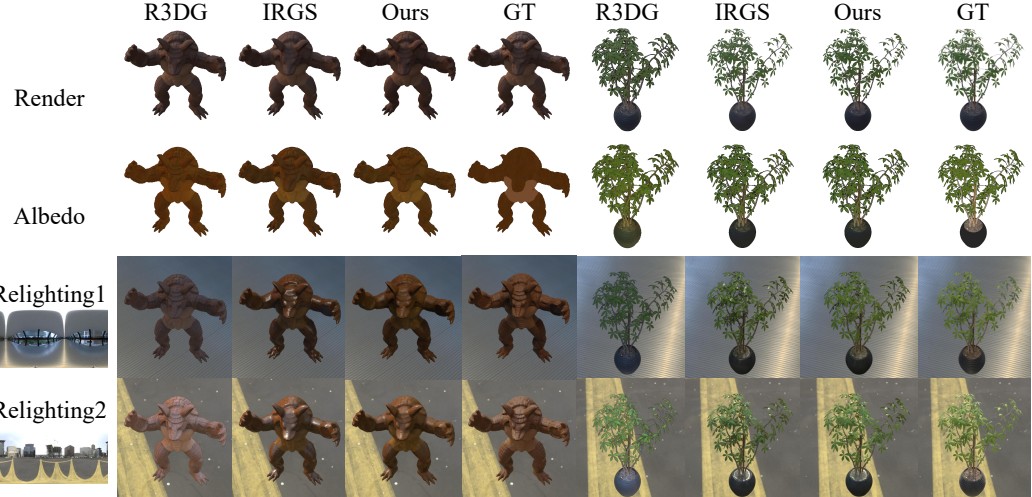

Figure 3: Qualitative comparison of NVS, albedo, and relighting results on the TensoIR dataset.

Gaussian ray tracing remains the dominant bottleneck. We address this by substituting 2DGRT with mesh-based ray tracing during relighting. Our approach uses truncated signed distance fusion (TSDF) Zhou et al. (2018) to extract triangle meshes while storing material attributes on mesh vertices, thereby simplifying alpha-blended materials to direct queries of the first-intersected face's attributes. This reduces intersection complexity from hundreds of ray-splat tests to a single ray-triangle intersection, achieving magnitude-order acceleration. We retain 2DGRT during training to fully exploit the learnable $c_i$ for precise indirect illumination. Experimental results confirm negligible quality loss when transitioning between Gaussian and mesh-based ray tracing.

## 3.5 TRAINING SCHEME

To accurately model complex light-surface interactions through geometry-sensitive ray tracing, establishing reliable geometry proves essential. Following the common practice Gu et al. (2025a), we implement a two-stage training process. The first stage employs Ref-Gaussian Yao et al. (2025), which utilizes physically-based deferred rendering with split-sum approximation, to reconstruct high-fidelity geometry for both low-gloss and glossy surfaces. Compared to IRGS's Gu et al. (2025a) 2DGS-based geometry initialization, our approach demonstrates superior capability in handling glossy materials. The second stage concentrates on material and lighting estimation as detailed in Section 3.2. During training, we selectively evaluate the rendering equation on a subset of pixels per viewpoint, significantly reducing computational overhead, and the denoiser is only used during relighting. We adopt a similar loss function as in IRGS:

$$\mathcal{L} = \mathcal{L}_c + \lambda_1^{pbr}\mathcal{L}_1^{pbr} + \lambda_{light}\mathcal{L}_{light} + \lambda_{s,a}\mathcal{L}_{s,a} + \lambda_{s,r}\mathcal{L}_{s,r} + \lambda_{s,m}\mathcal{L}_{s,m}, \quad (8)$$

where $\mathcal{L}_c$ represents the reconstruction loss Kerbl et al. (2023) for the outgoing radiance $\mathcal{C}$, $\mathcal{L}_1^{pbr}$ denotes the L1 loss between the final physically rendered pixels and GT, $\mathcal{L}_{light}$ regularizes incident illumination toward natural white balance, and $\{\mathcal{L}_{s,a}, \mathcal{L}_{s,r}, \mathcal{L}_{s,m}\}$ impose edge-aware smoothness constraints Gu et al. (2025a) on pixel-level albedo, roughness, and metallic maps respectively.

Table 2: Quantitative comparison of relighting results on GlossySynthetic dataset.

| | NeRF-based | | | 3DGS-based | | | | | |
| | Nvdiffrec-MC | TensoSDF | NeRO | R3DG | GS-ROR² | Ref-Gaussian | IRGS | Ours ($N_r = 512$) | Ours ($N_r = 16$) |
| | PSNR/SSIM | PSNR/SSIM | PSNR/SSIM | PSNR/SSIM | PSNR/SSIM | PSNR/SSIM | PSNR/SSIM | PSNR/SSIM | PSNR/SSIM |
|---|---|---|---|---|---|---|---|---|---|
| Angel | 22.89/0.865 | 20.40/.8969 | 16.21/.7819 | 16.65/.8013 | 20.81/.8775 | 21.39/0.9003 | 20.58/0.8596 | 24.21/0.9068 | 24.15/0.9053 |
| Bell | 24.30/0.903 | 29.91/.9767 | 31.19/.9794 | 16.15/.8391 | 24.49/.9267 | 22.90/0.9197 | 20.98/0.8779 | 25.96/0.9335 | 25.81/0.9315 |
| Cat | 23.88/0.907 | 26.12/.9354 | 28.42/.9579 | 17.49/.8503 | 26.28/.9421 | 20.54/0.9119 | 22.43/0.8881 | 27.21/0.9398 | 27.10/0.9374 |
| Horse | 26.42/0.935 | 27.18/.9567 | 25.56/.9437 | 20.63/.8832 | 23.31/.9376 | 24.97/0.9441 | 22.10/0.9208 | 24.81/0.9419 | 24.80/0.9415 |
| Luyu | 23.60/0.859 | 19.91/.8825 | 26.22/.9092 | 17.47/.8168 | 22.61/.8995 | 19.74/0.8753 | 22.73/0.8523 | 25.74/0.8966 | 25.73/0.8955 |
| Potion | 22.07/0.858 | 27.71/.9422 | 30.14/.9561 | 14.99/.7799 | 25.67/.9175 | 20.06/0.8677 | 22.92/0.8663 | 27.55/0.9184 | 27.42/0.9155 |
| Tbell | 22.60/0.883 | 23.33/.9404 | 25.45/.9607 | 15.99/.7965 | 22.80/.9180 | 20.74/0.9038 | 19.97/0.8535 | 22.21/0.8918 | 22.07/0.8869 |
| Teapot | 22.45/0.899 | 25.16/.9482 | 29.87/.9755 | 17.36/.8389 | 21.17/.8932 | 21.78/0.9237 | 19.27/0.8699 | 23.58/0.9250 | 23.55/0.9241 |
| Mean | 23.53/0.889 | 24.97/.9349 | 26.63/.9331 | 17.09/.8258 | 23.39/.9140 | 21.51/0.9058 | 21.37/0.8736 | 25.16/0.9192 | 25.08/0.9172 |
| Training Time | 4h | 6h | 12h | 1h | 1.5h | 0.6h | 1h | 0.7h | 0.7h |
| Ren. Time (FPS) | 2.5 | 1/4 | 1/4 | 1.5 | 122 | 208 | 0.5 | 1.5 | 25 |

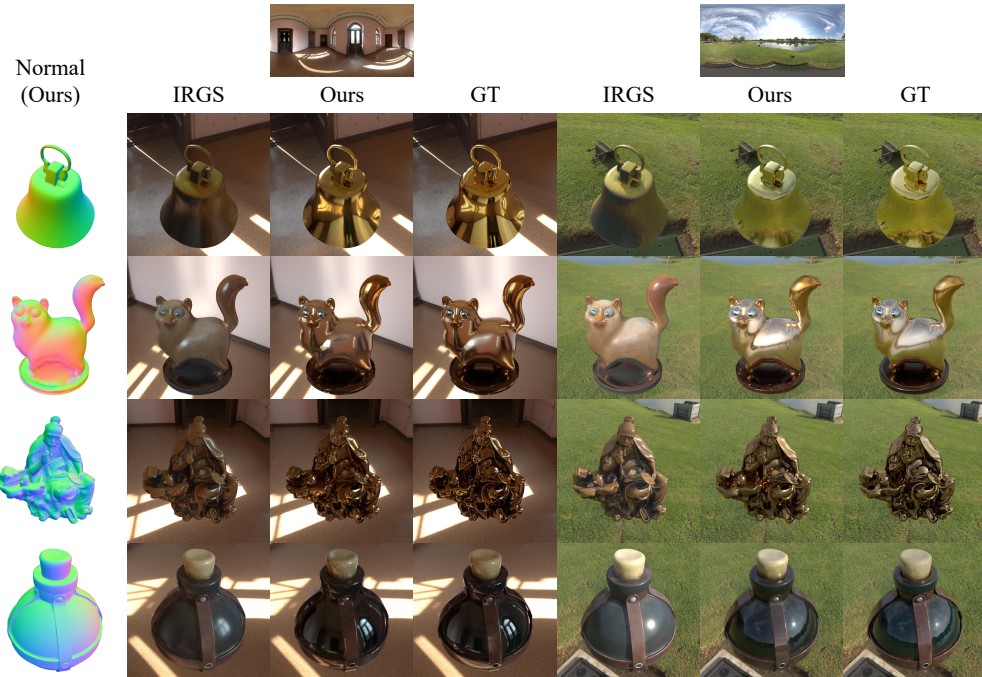

Figure 4: Qualitative comparison of relighting results on the GlossySynthetic dataset.

# 4 EXPERIMENT

**Datasets and metrics.** For quantitative evaluation, we utilize two synthetic datasets with ground truth material maps and relighting images: one low-gloss TensoIR datasets Jin et al. (2023) and one glossy GlossySynthetic dataset Liu et al. (2023). We employ PSNR, SSIM (Wang et al., 2004), and LPIPS (Zhang et al., 2018) to assess novel view synthesis, albedo, relighting, and estimated environment maps. For surface normal, we adopt mean angular error (MAE). We further conduct qualitative evaluations on three real-world datasets (RefReal Verbin et al. (2022), GlossyReal Liu et al. (2023), and Stanford-ORB Kuang et al. (2024) dataset).

**Implementation details.** Our training process consists of two stages. The first stage follows the original configurations of Ref-Gaussian Yao et al. (2025), while the second stage extends for an additional 10,000 iterations with loss weights consistent with IRGS Gu et al. (2025a). For MIS, we implement two configurations: a high-quality setting sampling 512 rays (256 cosine-weighted, 128 GGX, 128 light) and an efficient setting allocating 32 rays (16 cosine, 8 GGX, 8 light) for low-gloss dataset or 16 rays (8 cosine, 4 GGX, 4 light) for glossy dataset. We implement the mesh-based ray tracing in Optix Parker et al. (2010) via PyTorch CUDA extensions, utilizing meshes reconstructed from learned Gaussians via truncated signed distance fusion (TSDF). Material attributes are encoded per vertex in the extracted mesh. We employ $64 \times 128$ resolution environment maps for low-gloss materials, while glossy surfaces necessitate $128 \times 256$ resolution maps to adequately capture specular reflections. The complete training pipeline requires 40 minutes (30 minutes for the first stage, 10 minutes for the second stage), and the VRAM consumption is around 10 GB on a RTX 3090 GPU.

Table 3: Quantitative comparison of estimated environment maps on GlossySynthetic dataset.

|  | 3DGS-DR | GShader | Ref-Gaussian | IRGS | Ours |
|---|---|---|---|---|---|
| PSNR↑ | 9.04 | 6.52 | 14.70 | 7.18 | 19.49 |
| SSIM↑ | 0.435 | 0.320 | 0.599 | 0.304 | 0.616 |
| LPIPS↓ | 0.53 | 0.61 | 0.44 | 0.67 | 0.52 |

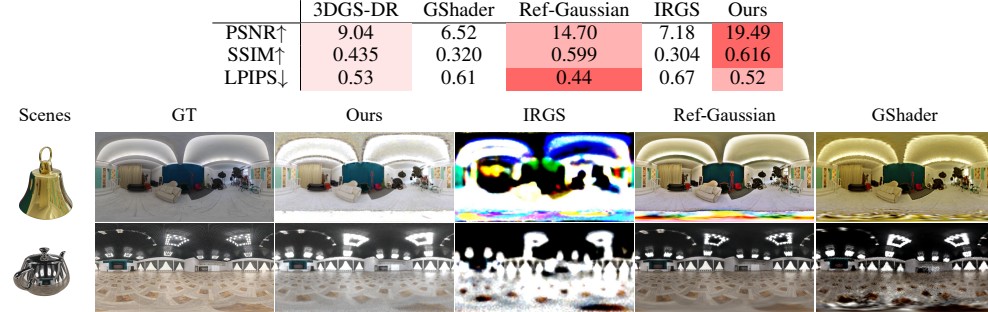

| Scenes | GT | Ours | IRGS | Ref-Gaussian | GShader |

Figure 5: Qualitative comparison of estimated environment maps on the GlossySynthetic dataset.

## 4.1 RESULTS ON SYNTHETIC DATA

**TensoIR.** In Table 1, we provide metrics for NVS, albedo estimation, and relighting results on TensoIR dataset Jin et al. (2023). Compared to previous arts Zhang et al. (2021; 2022); Jin et al. (2023); Liang et al. (2024); Gao et al. (2024); Zhu et al. (2024b); Gu et al. (2025a), our high-quality setting ($N_r = 512$) achieves state-of-the-art relighting quality, while our efficient setting ($N_r = 32$) has only 0.29dB degradation in PSNR and also outperform pervious arts. Figure 3 demonstrates qualitative comparisons between *IRGS++* and 3DGS-based competitors Gao et al. (2024); Gu et al. (2025a) through NVS, albedo, and relighting. *IRGS++* achieves photorealistic relighting with smooth results and accurate inter-reflections, whereas IRGS shows limited quality in specular modeling and R3DG completely fails to model secondary light effects.

**GlossySynthetic.** For GlossySynthetic dataset Liu et al. (2023), we evaluate *IRGS++*'s performance in relighting. As shown in Table 2, *IRGS++* achieves superior relighting quality among 3DGS-based methods, though NeRF-based approaches like TensoSDF Li et al. (2024) and NeRO Liu et al. (2023) exhibit higher metrics, partly attributed to their employment of Blender's physically-based path tracing for global illumination in relighting, as they do not native support global illumination with implicit field. For fairness, we evaluate IRGS Gu et al. (2025a) with its first stage substitute to Ref-Gaussian Yao et al. (2025), as its original 2DGS implementation fails to reconstruct the geometry. Figure 4 demonstrates *IRGS++*'s capacity for modeling accurate highlight regions, whereas IRGS exhibits substantial artifacts due to the lack of importance sampling and metallic. Table 3 gives quantitative evaluation of estimated environment maps, where our method outperforms on most metrics. Figure 5 illustrates our most natural estimations. Note that although Ref-Gaussian appears smooth, its relighting and environment maps score poorly because it leverages spherical harmonics to model the diffuse term, causing incorrect albedo estimation.

## 4.2 RESULTS ON REAL-WORLD DATA

In Figure 6, we conduct experiments on real-world datasets, including the RefReal dataset Verbin et al. (2022), and GlossyReal dataset Liu et al. (2023). Due to the lack of ground-truth material maps and relighting results, we provide qualitative results. To mitigate the impact of unbounded geometry on relighting quality, we only consider regions within a predefined spherical boundary, thereby enabling more effective ray tracing. In Figure 6, the reflective sphere in the "gardenspheres" exhibits precise reflections of novel lighting and inter-reflections from adjacent objects, confirming *IRGS++*'s capability to handle specular effects in complex scenarios. We also provide visualizations on Stanford-ORB dataset Kuang et al. (2024) in Figure 15, please refer to supplementary.

## 4.3 ABLATION STUDY

Figure 8 presents ablation studies analyzing critical components of *IRGS++*: "w/o denoiser" (direct output of Monte Carlo estimator), "w/o MIS" (stratified sampling replacing multiple importance sampling), and "w/o mesh" (persisting with 2D Gaussian ray tracing during relighting). The full model maintains stable rendering quality when reducing ray samples from 512 to 16, while "w/o denoiser" and" w/o MIS" exhibit significant degradation under sparse sampling. Notably, the "w/o mesh" variant achieves comparable quality to the full model, confirming mesh-based ray tracing pre-

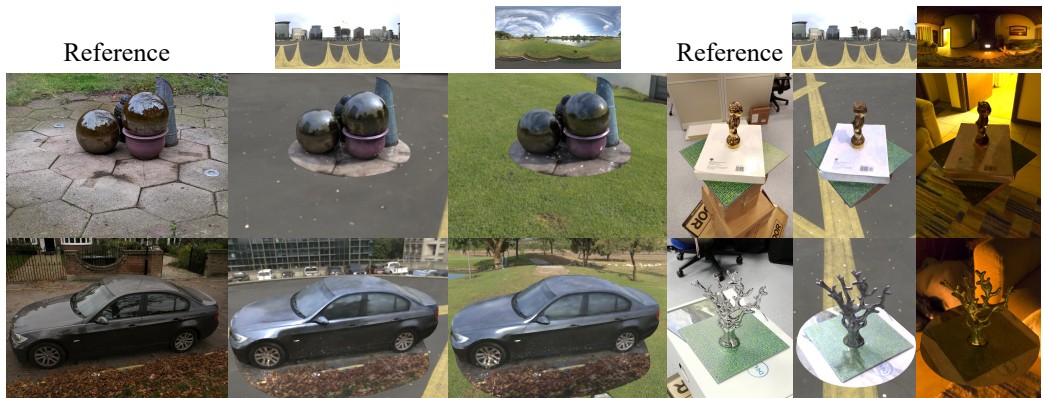

Figure 6: Relighting results on the real-world RefReal dataset and GlossyReal dataset.

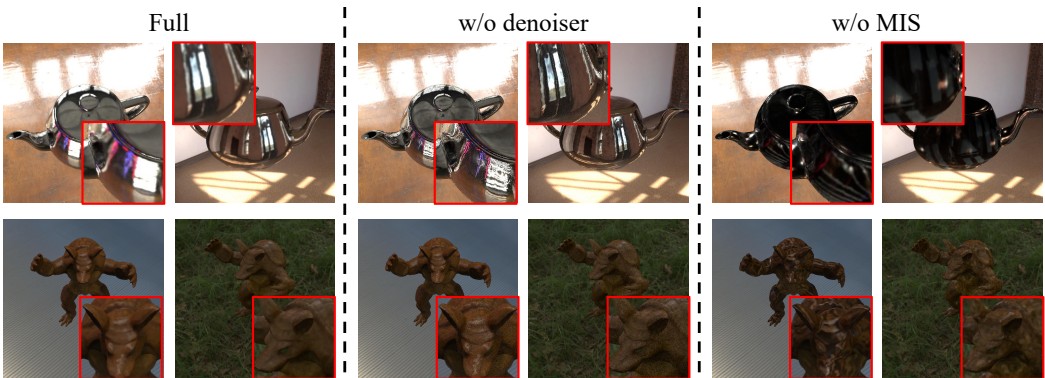

Figure 7: Ablation studies on denoiser and multiple importance sampling (MIS).

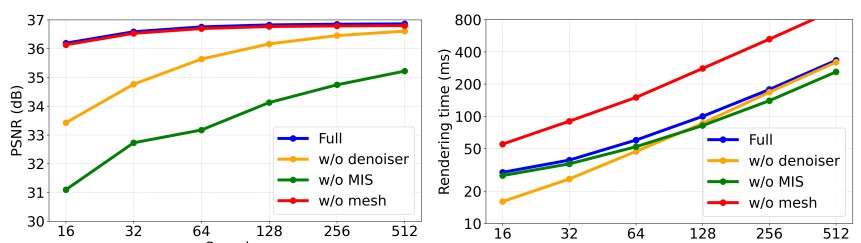

Figure 8: Ablation studies on various components of *IRGS++* using "Armadillo" scene.

serves accuracy while significantly accelerating computation. The narrow gap between "full" and "w/o denoiser" is only significant under sparse sampling, attributed to cross-bilateral filter overhead. Meanwhile, "w/o MIS" marginally outperforms "full", this is because the MIS is more complex than stratified sampling. Figure 7 qualitatively demonstrates these effects: "w/o denoiser" exhibits pronounced Monte Carlo noise under sparse sampling, while "w/o MIS" produces inaccurate specular highlights because it misses strong directional illumination with too few samples.

## 5 CONCLUSION

In this paper, we introduce *IRGS++*, a novel framework that achieves photorealistic secondary light effects spanning from diffuse to specular surfaces, while achieving significant acceleration compared to IRGS. *IRGS++* employs multiple importance sampling (cosine, GGX, and light sampling) to reduce ray sampling in Monte Carlo integration while better capturing light effects. A cross-bilateral filter is also applied to the Monte Carlo estimator to further eliminate noise under limited sampled rays. Furthermore, we replace 2D Gaussian ray tracing with mesh-based ray tracing during relighting, reducing computational complexity from hundreds of ray-splat intersection queries to a single ray-triangle intersection per ray. Extensive experiments across both low-gloss and glossy material datasets demonstrate the superior rendering quality and efficiency of *IRGS++*.

**Ethics Statement**    This work follows the ICLR Code of Ethics. Our research is focused on computer graphics and inverse rendering, aiming to improve the efficiency and accuracy of modeling light interactions in 3D scenes. The study does not involve human subjects, personal or sensitive data, or applications that could directly cause harm. The datasets used are publicly available and widely adopted in the research community, and we have carefully documented their use to ensure transparency and integrity. We do not foresee ethical risks related to fairness, privacy, or potential misuse of the proposed method.

**Reproducibility Statement**    We have taken deliberate steps to make our work reproducible. The proposed method is described with sufficient detail for implementation, including how we accelerate light transport modeling and reduce noise in rendering. All datasets employed are publicly available. We provide experimental details and ablation results to clarify design choices.

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

## A    MORE DETAILS

### A.1    MESH CONVERSION

After the getting the robust geometry from the first stage, we utilize truncated signed distance fusion (TSDF) to extract meshes using Open3D Zhou et al. (2018). We first render the depth map using Gaussian splatting, then project the depth map onto a voxel grid to convert it into truncated signed distances from the object surface to the voxel center. Finally, the marching cubes algorithm is applied to extract the zero-level isosurface, which is then converted into a triangle mesh. The whole mesh conversion process completes in less than 20 seconds.

## B    MORE RESULTS

A supplementary video demonstrating full 360-degree perspective visualization of the scenes presented within the paper is included in the supplemental materials.

### B.1    RESULTS ON A COMPOSITED SCENE

In Figure 9, we present a relit scene comprising reconstructed objects from the TensoIR Jin et al. (2023) and GlossySynthetic Liu et al. (2023) datasets, demonstrating *IRGS++*'s capacity to concurrently process both low-gloss and glossy objects. Visualizations of indirect illumination (considering only the indirect term in Eq. 4) and global illumination are provided for comprehensive analysis. In Figure 10, we provide the rendered material maps of the compsited scene. The scene is relighted using only 32 samples per pixel, yet achieves high-fidelity relighting results with physically plausible secondary effects, as evidenced by the accurate ground reflectance of the specular "bell".

### B.2 RESULTS ON TENSOIR DATASET

In Figure 11, we present a comprehensive qualitative comparison of two additional scenes from the TensoIR dataset Jin et al. (2023). Our method achieves comparable or superior material decomposition and relighting fidelity relative to IRGS Gu et al. (2025a), while simultaneously attaining significantly accelerated rendering speed. A qualitative comparison of rendered normal maps in Figure 12 further demonstrates the enhanced geometry quality enabled by our integration of Ref-Gaussian Yao et al. (2025) during the first stage.

### B.3 RESULTS ON GLOSSYSYNTHETIC DATASET

In Figure 13, we present an extended evaluation of relighting results across four additional scenes within the GlossySynthetic dataset Liu et al. (2023). *IRGS++* demonstrates superior relighting performance compared to IRGS Gu et al. (2025a), achieving physically accurate specular reflectance. The corresponding normal maps exhibit high fidelity, which is attributed to the robust geometry reconstruction during the initial stage.

### B.4 RESULTS ON STANFORD-ORB DATASET

In Figure 15, we present an visualization of the relighting results on Standford-ORB dataset Kuang et al. (2024), demonstrating our method's capability on diverse real-world data.

## C DISCUSSIONS ON THE POTENTIAL SOCIAL IMPACTS

*IRGS++* leverages Gaussian splatting for more robust inverse rendering and accelerated global illumination, helping small teams create realistic 3D content for movies or AR apps. However, this could reduce demand for traditional lighting artists, requiring workers to adapt to new tools. Errors in input 3D models (like gaps or inaccuracies) might lead to wrong material estimates, impacting fields like digital museum projects. *IRGS++*'s quick lighting/material editing could also be misused to falsify details in virtual scenes, needing protective measures. We hope it encourages blending AI with physically-based rendering to improve digital replicas of real-world scenes.

## D THE USE OF LLM

Large language models were only used to aid and polish the writing of this paper. They played no role in research ideation, algorithm design, experimental setup, or result analysis.

## E LIMITATION

Despite implementing multiple acceleration strategies, our approach exhibits higher computational demands compared to real-time 3DGS-based methods employing split-sum approximations that sacrifice rendering accuracy. This limitation primarily arises from the fundamental requirement of evaluating the rendering equation with at least 16 rays per pixel to maintain photometric fidelity.

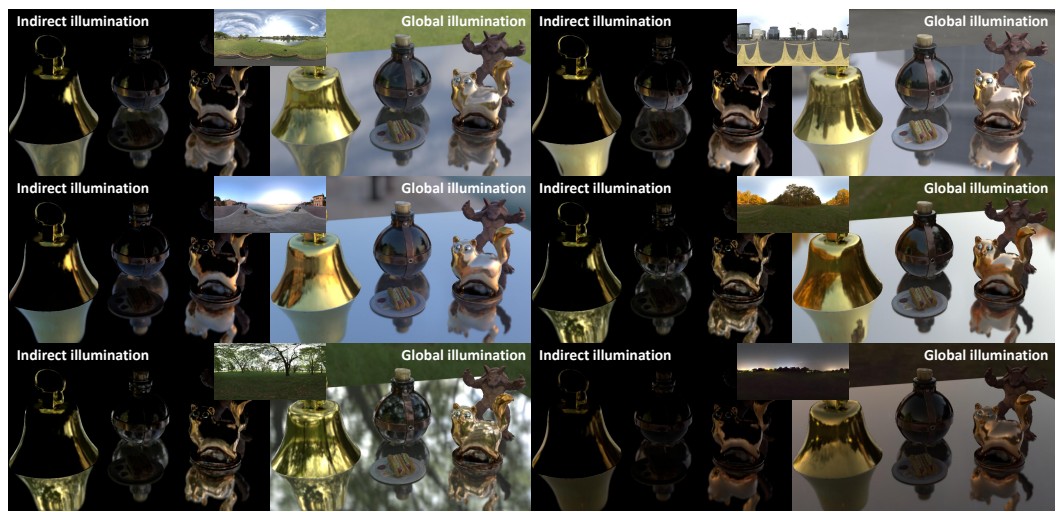

Figure 9: Indirect and global illumination in a composited scene using *IRGS++*.

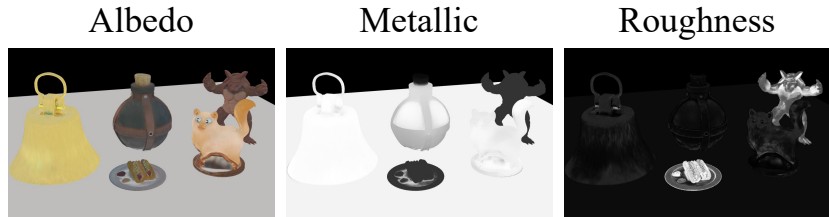

Figure 10: Estimated materials of the composited scene.

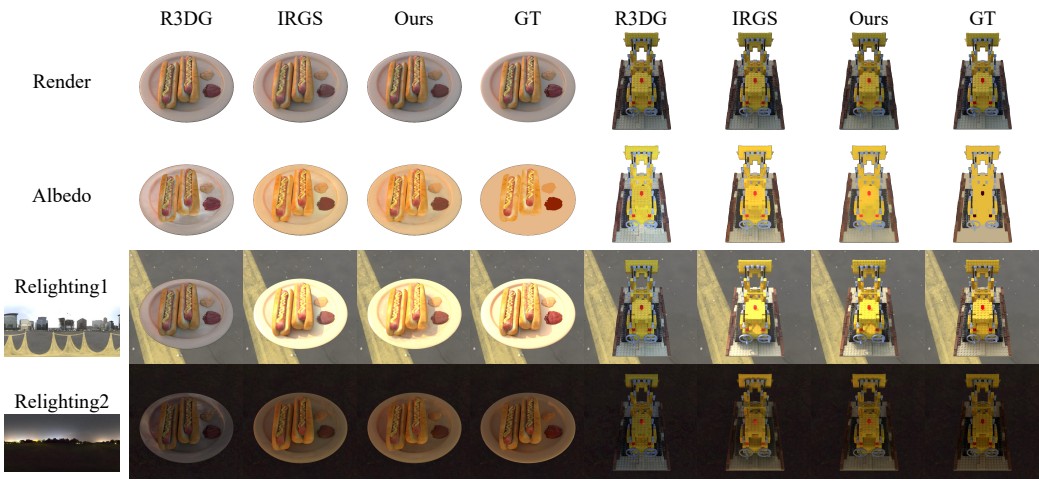

Figure 11: Qualitative comparison of NVS, albedo, and relighting results on the TensoIR dataset.

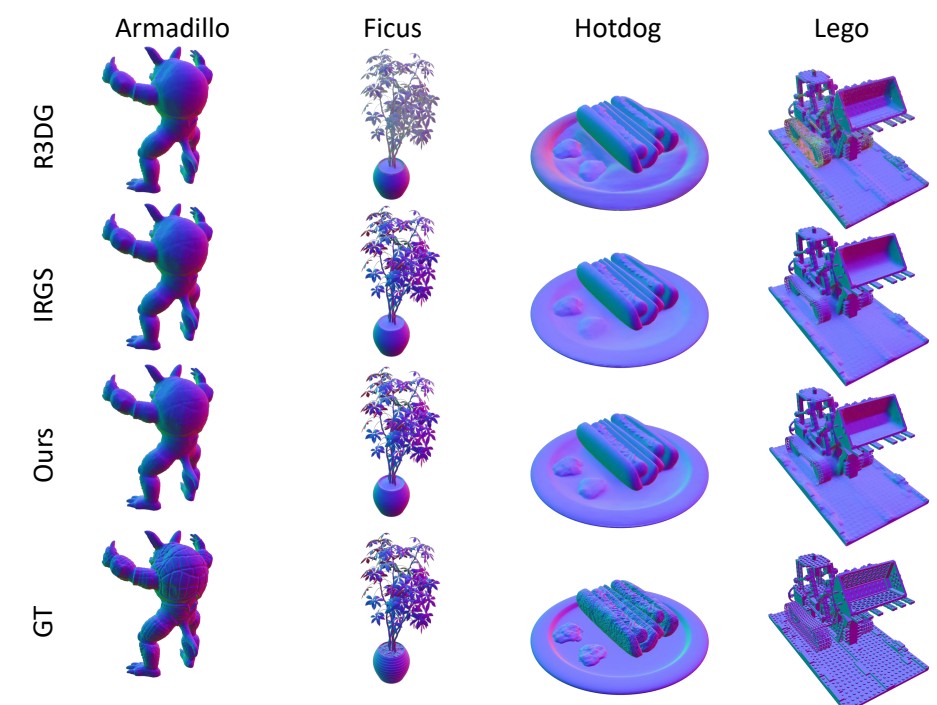

Figure 12: Qualitative comparison of rendered normal maps on the TensoIR dataset Jin et al. (2023).

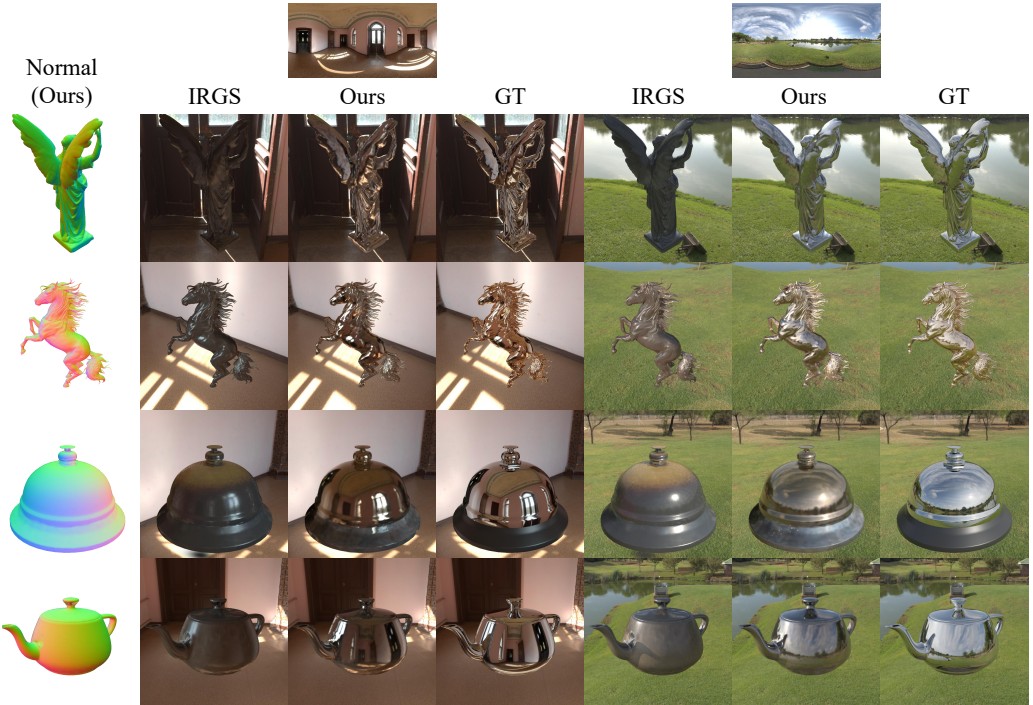

Figure 13: Qualitative comparison of relighting results on the GlossySynthetic dataset.

| Albedo | Metallic | Roughness | Diffuse | Specular | Normal |
|---|---|---|---|---|---|

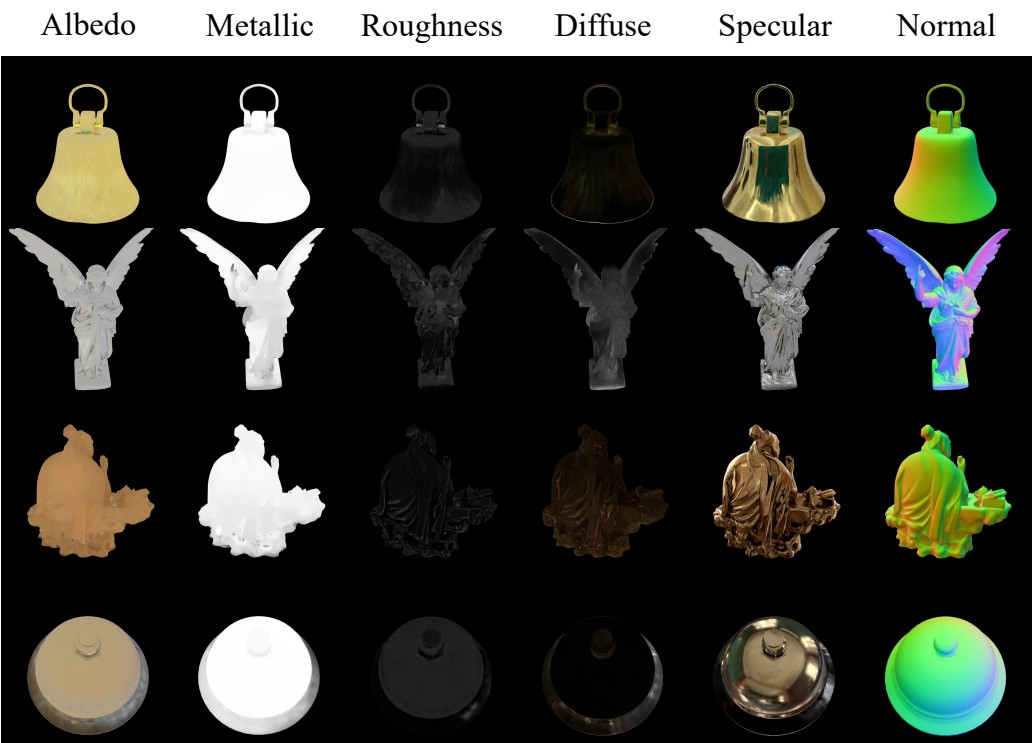

Figure 14: Qualitative comparison of estimated material maps on the GlossySynthetic dataset.

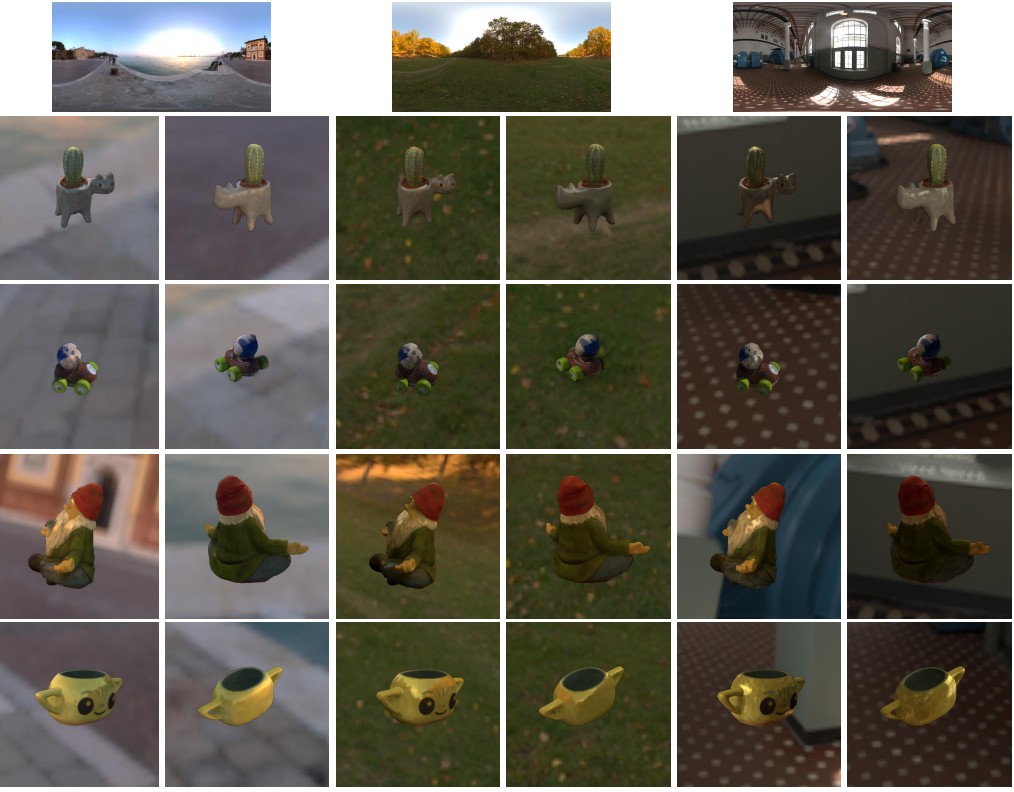

Figure 15: Relighting results on the real-world Stanford-ORB dataset.