# OpenReview forum: "IRGS++: Robust Geometry, Material, and Light Decomposition with Accelerated Inter-Reflective Gaussian Splatting"
_ICLR.cc/2026/Conference — ICLR 2026 Conference Withdrawn Submission_

### Official Review · Reviewer_FzvB · 2025-10-30

**Soundness:** 2
**Presentation:** 2
**Contribution:** 2
**Rating:** 4
**Confidence:** 5

**Summary:**

This paper presents a novel inverse rendering framework, IRGS++, which introduces MIS and denoiser to reduce the variance and mesh-based ray tracing to improve the efficiency. The proposed inverse rendering method extends the prior work IRGS with variance reduction strategies to better cover the circumstance of glossy surface and utilizes mesh-based ray tracing to replace the expensive 2DGRT during relighting.  Experiment results are provided to demonstrate the effectiveness of the framework, highlighting the good performance and efficiency of the proposed method.

**Strengths:**

1. This paper proposes a novel inverse rendering framework. By combining multiple importance sampling with a differentiable denoiser, achieve better material and lighting estimation than baselines while maintain the efficiency.
2. The proposed framework is applicable to both specular and diffuse surfaces.
3. Extensive experiments show that the method achieve better performance than IRGS.

**Weaknesses:**

1. The proposed methods may be too technical and lack of novelty. Since the MIS and denoiser are not new strategies to reduce the variance, if someone wants to enhance the rendering or inverse rendering performance with the same samples, they will easily think of using the techniques mentioned above, and these techniques are not difficult to implement. The same strategies have been used in previous work Nvdiffrecmc. And integrating mesh-based ray tracing in inverse rendering framework was also raised in previous work MIRReS.
2. The performance gain is not good enough. As shown in table 1, the NVS and albedo estimation performance of IRGS++ is comparable with IRGS. The relighting performance of IRGS++ is better than IRGS, but it seems strange since the albedo estimation results of both methods are very close and Figure 8 shows that the performance of 2DGRT and mesh-based ray tracing is almost the same. So where is the performance gain comes from? And for glossy surface circumstance, the author only give the relighting performance, while the proposed methods is inferior to NeRO on quality.

**Questions:**

1. As the proposed method can be seen as the extension of IRGS. Could you provide the performance using IRGS’s initialization instead of Ref-Gaussian’s, especially on Glossy dataset. This may better demonstrate the effectiveness of the variance reduction schemes and mesh-based RT.
2. In table 2, why $N_r = 512$ and $N_r = 16$ take the same training time?
3. Could you provide the indirect illumination visualization on TensoIR or Synthetic4Relight dataset？
4. Could you provide the NVS and material estimation comparison results on Glossy dataset?

---

### Official Review · Reviewer_R6DP · 2025-10-30

**Soundness:** 3
**Presentation:** 3
**Contribution:** 2
**Rating:** 6
**Confidence:** 4

**Summary:**

This work proposes a 3DGS-based inverse rendering framework building upon IRGS (CVPR'25) by incorporating several key techniques to address limitations in modeling glossy surfaces and rendering speed. Specifically, it introduces: (1) a metallic material parameter; (2) Multiple Importance Sampling (MIS) with cosine, GGX, and light sampling to reduce variance; (3) a cross-bilateral denoiser; and (4) a switch from Gaussian-based ray tracing to mesh-based ray tracing during the relighting phase for acceleration. Experiments on synthetic and real datasets demonstrate improved relighting quality on glossy objects and significantly faster relighting speeds compared to standard IRGS.

**Strengths:**

The method is well-motivated, and the experiments are solid to validate the effectiveness of the proposed methods. By incorporating a metallic parameter and MIS, the method handles glossy materials significantly better than the original IRGS. Further, achieving real-time or near real-time relighting (25 FPS claimed at lower sample counts) is a valuable practical improvement over the slow relighting of pure Gaussian ray tracing methods.

**Weaknesses:**

### 1. Limited Methodological Novelty

The core contributions largely comprise integrating well-established graphics techniques—such as MIS, denoising, and mesh extraction for ray tracing—into the existing IRGS framework. While effective, this represents an engineering combination of multiple prior works (e.g., Nvdiffrec-MC [1] for MIS/denoising, IRGS [2] as the base, and Ref-Gaussian [3] for initialization) rather than a fundamental methodological breakthrough. The paper would benefit from a deeper discussion justifying the specific choices of these components over alternatives.

---

### 2. Unclear Mesh-based Ray Tracing Details

Section 3.4 seems to lack sufficient detail regarding the mesh conversion and attribute assignment.

- It is unclear if there is a specific mesh-Gaussian binding process similar to recent works like [4]. Could you clarify the detailed process in Section 3.4 regarding how material attributes are assigned to mesh vertices? Is there an explicit mesh-Gaussian binding process involved (similar to [4])?
- Does the proposed mesh-based ray tracing support multi-bounce indirect lighting, or is it limited to single-bounce? How does this compare to other recent mesh-based inverse rendering works that employ multi-bounce path tracing (e.g., [5])?

Including [4,5] in the discussions could be beneficial.

---

### 3. Citation Format

Using `\cite` instead of `\citep` may reduce text clarity in some web browsers.

---

### References

[1] Shape, Light, and Material Decomposition from Images using Monte Carlo Rendering and Denoising (NIPS'22)

[2] IRGS: Inter-Reflective Gaussian Splatting with 2D Gaussian Ray Tracing (CVPR'25)

[3] Reflective Gaussian Splatting (ICLR'25)

[4] GeoSplatting: Towards Geometry Guided Gaussian Splatting for Physically-based Inverse Rendering (ICCV'25)

[5] Inverse Rendering using Multi-Bounce Path Tracing and Reservoir Sampling (ICLR'25)

**Questions:**

- What is the trade-off in the choice of Ref-Gaussian initialization instead of IRGS's (L.089-L.090)? Is it free lunch, or will it inherit shortcomings from Ref-Gaussian?
- In Section 3.4, is the extracted mesh used during relighting capable of producing high-quality renderings on its own (as a standalone mesh representation), or does it serve merely as a coarse proxy for ray intersection (like in [4])?

---

### Official Review · Reviewer_JSf8 · 2025-10-31

**Soundness:** 3
**Presentation:** 2
**Contribution:** 2
**Rating:** 2
**Confidence:** 5

**Summary:**

This paper proposes IRGS++, a Gaussian-splatting-based inverse rendering framework that captures inter-reflections for global illumination and glossy appearance. The pipeline is built upon the 2D Gaussian Raytracing (2DGRT) from IRGS and improves the rendering equation estimator from stratified sampling to Monte Carlo importance sampling. To reduce Monte Carlo noise, the paper leverages multiple importance sampling and a bilateral-filter-based denoiser. Results demonstrate superior performance over baselines.

**Strengths:**

The overall design is physically based and reasonable. Compared to the stratified sampling in IRGS, the Monte Carlo importance sampling estimator has better capability to capture high-frequency details, such as glossy reflections, which can be validated in Fig. 4's comparison with IRGS. The MIS and denoising strategies are widely used techniques to reduce Monte Carlo noise in computer graphics.

**Weaknesses:**

1. Incremental work with limited novelty. The proposed pipeline is basically a combination of existing methods. The 2DGRT algorithm is based on IRGS, the geometry is initialized by Ref-Gaussian, the MIS and denoising techniques are proposed by NVdiffrec-MC. I can't see any contribution with significant innovation.
2. Limited comparison. The paper only compares with GS-based and NeRF-based methods, while there is also a rich literature of mesh-based inverse rendering methods, such as MIRRES [1]. Besides, in Tab. 2, NVdiffrec-MC should be classified as mesh-based rather than NeRF-based.
3. Unclear paper writing. The proposed method should be a 2-stage pipeline, where the first stage is using Ref-Gaussian to initialize the geometry and the second is the inverse rendering stage with raytracing. This information is only mentioned in Fig. 2 and the "implementation detail" paragraph in Sec. 3.5. I believe that this is important information about the pipeline and should be emphasized in an individual paragraph.
4. Missing citations:

[1] A recent mesh-based inverse rendering pipeline:
```
@inproceedings{dai2025inverse,
  title={Inverse Rendering using Multi-Bounce Path Tracing and Reservoir Sampling},
  author={Dai, Yuxin and Wang, Qi and Zhu, Jingsen and Xi, Dianbing and Huo, Yuchi and Qian, Chen and He, Ying},
  booktitle={The Thirteenth International Conference on Learning Representations},
  year = {2025},
  url = {https://openreview.net/forum?id=KEXoZxTwbr}
}
```
[2] A 3DGS-based method enabling tracing secondary rays by an Unscented Transform rather than raytracing (See also questions)
```
@inproceedings{wu20253dgut,
  title={3dgut: Enabling distorted cameras and secondary rays in gaussian splatting},
  author={Wu, Qi and Esturo, Janick Martinez and Mirzaei, Ashkan and Moenne-Loccoz, Nicolas and Gojcic, Zan},
  booktitle={Proceedings of the Computer Vision and Pattern Recognition Conference},
  pages={26036--26046},
  year={2025}
}
```

**Questions:**

1. Please address my concerns in the "Weaknesses" section
2. The 3DGUT paper proposes a splatting-based technique for secondary ray query, which claims to be significantly more efficient than explicit raytracing. Can you provide a discussion about this, and whether the 2DGRT can be replaced by a "2DGUT"?

---

### Official Review · Reviewer_G6rP · 2025-11-02

**Soundness:** 2
**Presentation:** 2
**Contribution:** 1
**Rating:** 2
**Confidence:** 4

**Summary:**

This paper integrates Multiplexed Importance Sampling (MIS) into the IRGS framework to improve the efficiency of both inverse rendering and relighting. Additionally, the authors employ a mesh as a proxy during the relighting stage and incorporate cross-bilateral filtering to reduce variance, thereby significantly accelerating the relighting process.

**Strengths:**

The incorporation of MIS effectively accelerates the training process. Moreover, the use of mesh extraction as a proxy and the application of joint bilateral filtering for denoising significantly improve the frame rate during relighting.

**Weaknesses:**

Both MIS and joint bilateral filtering are standard techniques commonly used to reduce Monte Carlo variance. Additionally, extracting mesh representations from 2D Gaussian splats is a widely adopted practice.

The structure of Section 3 is somewhat unbalanced. A substantial portion is devoted to reiterating background information and referencing prior work (e.g., Sections 3.1-3.3), while the sections that describe the novel contributions and technical details are relatively brief and underdeveloped. This makes it difficult for readers to fully understand and assess the key implementation aspects of the proposed method.

Section 4.1 is somewhat difficult to follow due to the dense listing of baseline methods. This hinders readability and makes the narrative feel cluttered. Using pronouns when not referring to a specific baseline could improve the clarity and flow of this section.

**Questions:**

1. The loss terms presented in Equation (8) are insufficiently explained. In particular, the formulation and implementation details of the lighting loss, denoted as $L_{light}$, deserve further clarification.
2. There appears to be an inherent gap between using a mesh proxy and 2D Gaussian splats for relighting, as the training pipeline does not involve any mesh representation. Intuitively, introducing the mesh should introduce some approximation error or artifacts due to the domain mismatch. However, Figure 8 suggests that omitting the mesh proxy can mildly degrade rendering quality. Could the authors clarify why this counterintuitive result occurs?

---

### Note · Authors · 2025-11-26

I have read and agree with the venue's withdrawal policy on behalf of myself and my co-authors.